# Evaluation of the Effectiveness of Botulinum Therapy Based on the Anthropometric Characteristics of the Face Using Non-Invasive Thermal Imaging Data

**DOI:** 10.3390/diagnostics15192519

**Published:** 2025-10-04

**Authors:** Olesya Kytko, Yuriy Vasil’ev, Ekaterina Emelyanova, Evgeniy Kutin, Ramin Sarmadian, Sofia Trofimova, Irina Kondrina, Alexander Moiseenko, Sergey Dydykin, Ekaterina Rebrova

**Affiliations:** 1Sechenov First Moscow State Medical University of the Ministry of Health of the Russian Federation (Sechenov University), 119048 Moscow, Russia; kytko_o_v@staff.sechenov.ru (O.K.); i@ekutin.ru (E.K.); sarmadianr7@gmail.com (R.S.); sofia-trofimov-a@yandex.ru (S.T.); moiseenko_a_a@student.sechenov.ru (A.M.); dydykin_s_s@staff.sechenov.ru (S.D.); rebrova_e_v@staff.sechenov.ru (E.R.); 2Antimicrobial Photodynamic Therapy Lab, MIREA—Russian Technological University, 125993 Moscow, Russia; 3LLC “U Clinic”, 400074 Volgograd, Russia; deryabina.ekaterina.86@mail.ru; 4LLC Beauty and Health Center “ETIKA”, 129344 Moscow, Russia; etika_krasota@mail.ru

**Keywords:** thermography, subcutaneous fat tissue, botulinum neuroprotein, injections

## Abstract

**Objective**: The objective of this study was to clarify the connection between BTX-A injections and local changes in skin temperature and to assess the correlation between post-BTX-A injection facial vascular hyperthermia and subcutaneous adipose tissue thickness (SAT) in the frontal area using thermography. **Methods**: The study involved 30 patients (mean age 42 ± 0.5 years; 18 women, 12 men). Facial skin temperature was measured via thermography (Thermo GEAR G30) before, immediately after, and 20 min after subcutaneous injection of BTX-A with hemagglutinin complex, gelatin (6 mg), and maltose monohydrate (12 mg). SAT development was graded by combined visual-palpation assessment. Statistical analysis included Student’s *t*-test and the Mann–Whitney U-test. **Results**: Biphasic thermal response: immediately post-injection: Significant decrease in min (−1.1 °C) and mean (−0.3 °C) facial temperatures (*p* < 0.05); 20 min post-injection: pronounced increase in mean (+1.5 °C), max (+1.3 °C), and min (+1.6 °C) temperatures (*p* < 0.001), attributed to BTX-A-induced vasodilation and local inflammation. Subjects with pronounced SAT exhibited significantly higher baseline temperatures (Me = 33.1 °C vs. 29.8 °C; *p* < 0.001) and more intense hyperthermic responses (+1.6 °C mean increase vs. +1.1 °C in low-SAT group; *p* < 0.001). Pronounced SAT was predominantly female (10/15; *p* < 0.05) and linked to higher BMI (33.3% overweight vs. 0% in low-SAT; **p* = 0.036*). **Conclusions**: SAT thickness is a key determinant of post-BTX-A vascular hyperthermia, with pronounced SAT predicting stronger reactions. **Practical Recommendation**: Targeted local hypothermia (+4 °C to +8 °C for 5–7 min post-injection, adjustable by SAT thickness) mitigates hyperemia, edema, hematoma risk, and potential toxin diffusion, especially in high-SAT individuals.

## 1. Introduction

In recent years, botulinum neuroprotein type A (BTX-A) has become a leading pharmacological agent in esthetic medicine. Its widespread use is attributed to its proven efficacy in correcting age-related skin changes, as well as its ability to deliver a visible rejuvenating effect without surgical intervention. The statistically significant increase in the use of botulinum toxin therapy in cosmetic practice is associated with its pronounced muscle-relaxing effect. Injections of BTX-A enable the smoothing of dynamic wrinkles and improvement of skin texture architecture. According to the International Society of Aesthetic Plastic Surgery (ISAPS), over 8.8 million cosmetic botulinum toxin procedures were performed in 2023 alone [1].

However, despite the apparent technical simplicity of the procedure and its high safety profile when performed correctly, BTX-A injections may be accompanied by complications. One adverse effect is facial asymmetry—a condition characterized by disproportionate changes in muscle tone, which can negatively impact appearance and cause psychological distress in patients. After injection, BTX-A molecules can diffuse into adjacent muscle structures. Connective tissue partitions between muscles do not serve as an insurmountable barrier to this process [2]. The key mechanism of diffusion is Brownian motion, the speed of which directly depends on temperature. A local increase in temperature potentially enhances the kinetic activity of BTX-A molecules, increasing the likelihood of their spread beyond the target zone. Given that, the diffusion can lead to an asymmetric distribution of the drug in the right and left halves of the face. As a result, the neurotoxin can affect different muscle groups, which can eventually manifest as muscle asymmetry. Additionally, improperly administered botulinum toxin therapy may lead to edema.

The human body maintains a relatively stable core temperature of approximately 37.5 °C in the homeothermic core, which includes internal organs (abdominal cavity), the brain, heart, blood in major arteries, and deep muscles. The temperature of peripheral tissues is lower and can vary widely. The boundary between the homeothermic core and the poikilothermic shell is subject to changes depending on ambient temperature [3].

The primary factor determining local temperature is microcirculation intensity. Thus, local thermometry can be used to evaluate the effectiveness of various types of interventions, both topical and systemic [4].

Thermography is a non-invasive diagnostic method that measures body surface temperature. This technique is based on detecting infrared radiation emitted by the human body, which can be recorded by a specialized thermal imaging device and converted into a thermogram [5]. Thermography enables non-invasive assessment of tissue activity in the studied area and evaluation of tissue perfusion [4].

According to the literature [6,7], muscle contraction affects body temperature due to contractile thermogenesis, a mechanism of heat production associated with muscle contractions. With intramuscular administration of botulinum toxin, two effects develop: inhibition of extrafusal muscle fibers by inhibiting the nerve endings of alpha motor neurons at the level of the neuromuscular synapse, and inhibition of muscle spindle activity by inhibiting the gamma motor neuron cholinergic synapse on the intrafusal fiber. A decrease in gamma activity leads to relaxation of the intrafusal fibers of the muscle spindle and reduces the activity of Ia afferents. This leads to a decrease in the activity of muscle stretch receptors and the efferent activity of alpha and gamma motor neurons (clinically, this is manifested by pronounced muscle relaxation at the injection site and a significant decrease in pain).

The mechanism of the damaging effect of botulinum neurotoxin is the destruction of SNARE proteins in the presynaptic terminals of cholinergic nerves, which leads to a disruption of the release of acetylcholine into the synaptic cleft and the cessation of excitation transmission between neurons [8]. According to [9], acetylcholine, which normally causes endothelium-dependent vasodilation by inducing nitric oxide production in the presence of endothelial dysfunction, can also induce vasospasm. Thus, decreased synthesis and release of acetylcholine from endothelial cells leads to vasoconstriction initially. This may be further exacerbated in the short term by the fact that BTX-A injections into the frontal region are painful not only due to the anatomical features but also because of the slightly acidic solution administered.

Subcutaneous adipose tissue (SAT) is known to have pronounced heat-insulating properties [10]. We suggest that this insulating function may modulate the cutaneous vascular response to a pharmacological stimulus.

A review of domestic and international literature reveals a lack of studies investigating local skin temperature changes following BTX-A injections. The authors aimed to examine the effects of botulinum toxin therapy on facial skin temperature in the injection area using thermography and to develop a methodology for preventing positive thermal asymmetry.

BTX-A is widely used in esthetic and medical practice, not only for reducing wrinkles but also due to its vasomotor effects on skin perfusion. Previous studies suggest that BTX-A may influence blood flow, vessel diameter, and local skin temperature, with potential modulation by SAT [2,11]. However, the dynamics of these effects, particularly in regions with varying SAT levels, remain poorly understood.

Research Objective: The objective is to clarify the connection between BTX-A injections and local changes in skin temperature using infrared thermography, to assess the relationship between the hyperthermic response of facial vascular networks and the degree of subcutaneous adipose tissue thickness in the frontal region using thermography.

## 2. Results

Immediately after administration of the injection, both the minimal and mean facial temperatures exhibited reductions of 1.1 °C and 0.3 °C, respectively. By 20 min following the procedure, the mean, peak, and minimal facial temperatures demonstrated increases of 1.5 °C, 1.3 °C, and 1.6 °C, respectively. This biphasic thermal response likely stems from BTX-A-induced vasodilation and localized secretion of inflammatory mediators (Figure 1).

The study revealed a direct correlation between the degree of facial subcutaneous adipose tissue (SAT) development and both the velocity and magnitude of vascular responses in the injection zone. The rate of vascular reaction is a time parameter that reflects how quickly the temperature changes after a functional test. And the magnitude of the vascular reaction is an amplitude parameter that shows the intensity of the temperature change caused by shifts in blood flow [11] (Table 1).

Thermographic analysis across three temporal phases—pre-injection, immediate post-injection, and 20 min post-injection—demonstrated statistically significant differential responses (*p* < 0.001) between cohorts with minimal versus pronounced subcutaneous adipose tissue (SAT) development (Figure 2).

Subjects with greater SAT development exhibited a more intense vascular response, resulting in a more pronounced increase in both mean and maximum facial skin temperatures (Figure 3).

A statistically significant association was found between gender and SAT development. Specifically, the “pronounced SAT” category predominantly comprised women (10 participants), while the “minimal SAT” group included participants of both genders in equal proportions (Table 2).

## 3. Discussion

Our study demonstrated a biphasic thermal response to BoNT-A injection, with an immediate decrease followed by a significant temperature increase at 20 min. Thermographic analysis revealed that subjects with more developed SAT exhibited faster and more intense vascular responses, and this group predominantly comprised women.

BTX-A causes vasodilation and increases vessel diameter in arteries by 40% and in veins by 46% compared with saline controls [12]. The application of type A botulinum toxin (BTX-A) can induce persistent changes in skin temperature, as recorded by infrared thermography (IRT). Practical data demonstrate that this effect exhibits a biphasic dynamic: a short-term temperature decrease (averaging 0.3 °C) in the first hours after injection, followed by a gradual increase (of 0.6 ° C) over the next 5 days. The observed changes reflect a complex mechanism of neurovascular interaction, involving initial presynaptic inhibition of the cholinergic synapse of the γ-motor neuron on the intrafusal fiber, followed by compensatory vasodilatory effects. The results confirm that IRT can objectively assess not only the severity but also the duration of functional microcirculatory changes following BTX-A injection [13].

The decrease in temperature directly reflects the decrease in muscle thermogenesis produced by continuously contracting muscles, since BTA blocks the release of acetylcholine from the motor neuron ending. We can say that our pilot study, with further expansion of the number of participants, substantiates an important additional physiological mechanism of rejuvenation in botulinum therapy: the skin in a state of muscle rest is cooled to normal temperature, which creates more favorable conditions for the preservation of existing collagen and the synthesis of new collagen. We can observe the normalization of blood flow and lymph flow in the skin capillaries due to muscle relaxation and improved skin quality.

The presence of intrafusal fibers in facial muscles, as confirmed by several studies [14,15,16,17], suggests that the basal activity of these muscles may be partly maintained by reflexive activation through proprioceptors. This physiological insight is relevant for understanding the mechanism of BTX-A, as injections target muscles involved in these reflex circuits, potentially contributing to the observed clinical effects.

The first medical application of BoNT was for the treatment of strabismus in the late 1970s [18]. Since then, its indications have expanded to hyperhidrosis, pain syndromes, spasticity, and cosmetic use. Carruthers JD and Carruthers JA first proposed the use of BTX-A for wrinkle reduction in 1992 [19], and its application has since expanded across multiple medical fields, including the treatment of strabismus, hyperhidrosis, neuropathic pain, and finger muscle spasticity, as well as enhancing the efficacy of standard therapy in Raynaud syn-drome [20].

Tolerance and absorption of toxins can indeed be reduced by hypothermia due to decreased blood flow and vascular permeability. Local cooling is accompanied by pronounced vasoconstriction, which is confirmed by experimental observations: already within the first minutes, skin perfusion can decrease by 45–77%. This effect is realized through the activation of several molecular mechanisms, including ROS- and TRPA1-dependent signaling pathways leading to vasoconstriction. Such vascular changes reduce not only regional blood flow, but also the potential of tissues to absorb various substances. This effect is confirmed by a decrease in intestinal absorption of L-dopa and urcil by approximately 40% with a decrease in temperature, as well as a decrease in vascular permeability of albumin-FITC under hypothermia in an experimental shock model. These data substantiate the assertion that hypothermia can limit the toxin load due to vasoconstriction and reduced absorption [21,22,23,24,25,26,27].

As described above, our study recorded an increase in facial temperature following botulinum therapy injections. Several mechanisms may explain how BTX-A leads to vessel lumen dilation and increased tissue perfusion, with the most plausible mechanism involving its effect on SNAP25 in the presynaptic membrane [20,28,29,30]. Following endocytic vesicle acidification mediated by the proton pump, the heavy chain (H-chain) of botulinum neurotoxin type A (BTX-A) undergoes conformational changes that enable its insertion into the vesicle membrane and the formation of a translocation channel. Through this channel, the light chain (L-chain) is unfolded and released into the cytosol. Once in the cytosol, the L-chain functions as a zinc-dependent metalloprotease that specifically cleaves nine amino acids from the C-terminus of the synaptosomal-associated protein SNAP-25 (residues 198–206), generating the truncated SNAP-25197 [28,30]. Intact SNAP-25 is required for synaptic vesicle docking, neurotransmitter release, and regulation of presynaptic calcium channels. Even partial cleavage of the SNAP-25 pool significantly disrupts exocytosis and thereby blocks synaptic transmission. As the intravesicular pH decreases due to the proton pump, the H-chain of BTX-A undergoes conformational rearrangements that promote its insertion into the endosomal membrane and formation of a translocation channel, facilitating the partial unfolding and subsequent translocation of the L-chain into the cytosol.

Once released, the L-chain of BTX-A acts as a metalloprotease, enzymatically cleaving several amino acids from the C-terminus of the SNAP25 receptor (SNAP25206), producing SNAP25197 [28,30]. Normally functioning SNAP25 is essential for vesicle attachment and subsequent neurotransmitter release, as well as for regulating presynaptic membrane calcium channels. Cleavage of just 2–3% of the total SNAP25 pool disrupts exocytosis, blocking nerve impulse transmission. The proteolytic product, SNAP25197, also directly inhibits exocytosis. Thus, BTX-A prevents neurotransmitter release from vesicles [28], including sympathetic neurotransmitters like adrenaline and noradrenaline. Reduced vasoconstrictive effects of sympathetic nerve endings on vascular smooth muscle cells lead to vasodilation and increased perfusion at the injection site. BTX-A may also promote vessel regeneration by modulating angiogenesis-regulating genes, thereby improving long-term tissue blood supply [28].

Kalandakanond S. et al. [31] investigated the mechanism of action of botulinum toxin serotype A in the mouse phrenic nerve–hemidiaphragm preparation, focusing on the relationship between proteolytic cleavage of the intracellular substrate SNAP-25 and toxin-induced paralysis. Immunoblot analysis demonstrated that loss of SNAP-25 immunoreactivity was ≤10% at 1 h after paralysis but exceeded 75% by 5 h. Competitive inhibition of toxin binding with Triticum vulgaris lectin, blockade of pH-dependent translocation with methylamine hydrochloride, and zinc chelation each prevented both paralysis and SNAP-25 cleavage. These findings indicate that SNAP-25 cleavage by serotype A fulfills the requirements of the multistep model of botulinum toxin action, encompassing receptor-mediated endocytosis, pH-dependent translocation, and zinc-dependent proteolysis. Moreover, the minimal SNAP-25 cleavage observed at 1 h suggests that inactivation of only a small but functionally critical pool of SNAP-25 is sufficient to induce paralysis. In another study [32], the relationship between BoNT/A-catalyzed cleavage of SNAP-25 and muscle function was investigated using the rat gastrocnemius muscle model. Cleaved SNAP-25 was detected in motor nerve terminals as early as 24 h after toxin injection and persisted for more than two months. Comparison of the ratios of cleaved to intact SNAP-25 throughout the time course—from the onset of BoNT/A-induced paralysis until functional recovery—revealed that paralysis was consistently associated with a ratio greater than 0.35 [32]. These findings suggest that the prolonged paralytic effect of BoNT/A is attributable to the sustained presence of cleaved SNAP-25, whereas recovery of muscle function depends on restoration of a critical balance between intact and cleaved SNAP-25.

A recent review summarized the pharmacology of BoNT/A in both systemic exposure (botulism) and local therapeutic use. The authors emphasized the key molecular mechanisms—uptake, distribution, receptor binding, translocation, and inhibition of neurotransmitter release—as well as the role of zinc metabolism and host sensitivity. Authors also highlighted that these mechanisms underlie BoNT/A’s clinically relevant effects, including its safety, efficacy, and potential analgesic, anticancer, and anti-inflammatory properties, thereby supporting its broader therapeutic potential [33].

In our work, we used botulinum toxin with the manufacturer’s declared value of 6.1. In the literature [34], there is data that the pain after injection is stronger when using unreconstituted botulinum toxin. This is explained in the work of Kranz G et al. (2006), where the authors write about the effect of an acidic solution on pain receptors [35].

Since arterioles are generally more susceptible to neural influences than venules, their dilation results in local arterial hyperemia. Increased tissue perfusion raises temperature due to heat from the blood and expands extracellular fluid volume.

This study identified a local increase in average, maximum, and minimum facial temperatures 20 min post-procedure. The authors suggest that this post-botulinum therapy effect may not be entirely favorable. One might assume that BTX-A injections reduce mechanical activity of facial muscles, thereby decreasing thermogenesis and tissue temperature in the area. However, both experimental data and theoretical rationale indicate the opposite. According to physical laws, an endogenous temperature rise at the injection site leads to more active and chaotic movement of drug molecules, potentially causing unexpected diffusion of BTXA into non-target muscles. This effect may be a key mechanism in the development of facial asymmetries (e.g., eyelid or lip corner ptosis).

This phenomenon can be explained not only due to the vasodilating effect of BTX-A. The injection itself, which involves penetration of a foreign body (the needle) into tissues, inevitably traumatizes cellular and tissue structures. The procedure area releases vasoactive substances such as pro-inflammatory cytokines and prostaglandins [36,37]. According to Stepanova T.V. A.N. et al. (2019), pro-inflammatory cytokines induce local vasodilation by affecting the vascular smooth muscle layer [34]. Prostaglandin release in the post-injection trauma zone rapidly triggers vascular reactions, further increasing temperature [38].

Short-term pain after injection is associated with mechanical tissue damage, but the persistence of the effect after 20 min is explained from the point of view of pathophysiology: the initial damage activates the synthesis of prostaglandins and other mediators, which continue to be produced and affect tissues for tens of minutes. This is evidenced by the study of Urakov A et al. (2023), which showed persistent thermal asymmetry against the background of the administration of various drugs [39]. In addition to the mentioned processes, the persistence of the effect 20 min after injection of botulinum toxin type A (BTA) may be associated, according to Sharova AA (2021), with its vegetative-modulating effect, since BTA selectively affects not only the cholinergic nerve endings of motor neurons, but also the sympathetic fibers innervating the vessels and sweat glands of the skin [38].

BTX-A acts by blocking the production of neurotransmitters involved in maintaining vascular tone. In a study by [11] showed that patients receiving BTX-A in the back experienced a temperature increase of 2° to 5 °C from day 0 to day 21 over 100% of the back area, as measured by thermography, that is, immediately from the day of injection.

The findings on primary vasodilation after botulinum neuroprotein injection support the use of localized hypothermia to stabilize the drug molecule in the target area. Current literature reports the development of thermostable botulinum neuroprotein molecules, though this is currently limited to type D [39], which prevents extrapolation to approved drugs.

Despite the apparent simplicity of diagnosing botulinum therapy efficacy via non-contact thermometry, few valid models or application experiences are described in the literature [40].

The rationale for ice application lies in the physiological effects of localized tissue cooling and hypoesthesia, both pre- and post-injection [41]. Low temperatures induce vasoconstriction, reducing blood flow in the intervention zone. This diminishes BTX-A delivery to nerve vasorum synapses, thereby mitigating its vasodilatory effects and the severity of local vascular reactions. Additionally, cold slows metabolic processes and substance diffusion in tissues, which is critical for limiting BTX-A spread beyond the target area and reducing risks of unintended muscle relaxation and other side effects. Thus, ice application before and after BTX-A injection may serve as a simple, accessible, and effective method to prevent pronounced hyperthermic vascular reactions and unwanted drug diffusion.

Based on the results obtained, we can say that immediately after BTX-A injection, apply a chilled gel mask for 2–3 min, especially in individuals with pronounced subcutaneous adipose tissue. Avoid cosmetic procedures and products that enhance vasodilation during the first 3 days after botulinum neuroprotein injections.

To minimize the risk of uncontrolled diffusion of the drug and local adverse effects (edema, hematomas, muscular asymmetry), localized cooling of the treated area is recommended immediately after injection. The optimal method involves applying a specialized cold mask or a clean ice pack (ice-generating gel) wrapped in sterile gauze or cloth to prevent skin frostbite. Cold application should be performed for 5–7 min immediately after injection, depending on the thickness of subcutaneous adipose tissue. The recommended temperature of the mask/ice pack should be approximately +4…+8 °C. Application should be discontinued if significant discomfort occurs.

For individuals prone to hematoma or edema formation, the procedure may be repeated 1–2 h after injection, but no more than 2–3 times within the first 24 h.

Cold applications should not remain in direct contact with bare skin for more than 3–5 min at a time; short cooling intervals with breaks are advised.

This protocol helps stabilize the injected drug in the target area, reduce swelling, prevent extensive hyperemia, and lower the risk of adverse effects.

## 4. Conclusions

This study demonstrated an objective increase in facial skin temperature following injections of botulinum toxin type A with hemagglutinin. This effect is unfavorable, as it may lead to unintended diffusion of the active substance into surrounding tissues, potentially resulting in ptosis and edema. In individuals with less pronounced subcutaneous adipose tissue, facial temperature changes were less marked, whereas those with thicker adipose tissue exhibited more intense vascular reactions. Local hypothermia prior to injection can reduce the severity of localized vascular responses and decrease the likelihood of adverse outcomes.

## 5. Materials and Methods

The study was conducted at the Department of Operative Surgery and Topographic Anatomy, Sechenov University, Moscow, Russia. Prior to the procedure, all participants provided written informed consent. Facial skin temperature measurements were performed using a Thermo GEAR G30 (entered into the State Register of Measuring Instruments of the Russian Federation, NEC Avio infrared Technologies Co., Ltd., Yokohama, Japan) thermal imaging camera (Russia; temperature range: −20 °C to +100 °C; resolution: 160 × 120 pixels; accuracy: ±0.2 °C). Subcutaneous injections of botulinum toxin type A (BTX-A) solution, combined with hemagglutinin complex and excipients (gelatin: 6 mg; maltose monohydrate: 12 mg), were administered by a qualified specialist into the designated facial regions under standardized conditions. Statistical analysis was conducted using Microsoft Office Excel 2016.

The study included 30 patients (18 women and 12 men) with a mean age of 42 ± 0.5 years. The sample size was calculated a priori using the GPower 3.1 software package. Based on the data from [39], where an effect of size d = [16] was found, a sample size of at least 28 people per group is required to achieve a statistical power of 80% with a significance level of α = 0.05. Thus, a sample of 30 people is sufficient to identify the expected effect. Participants were stratified into two groups based on the degree of facial subcutaneous adipose tissue (SAT) development: group 1 (n = 15): patients with minimal SAT, group 2 (n = 15): patients with pronounced SAT. Group allocation was determined through visual inspection and medical history assessment. Participants with chronic medical such as peripheral angio- and neuropathies (specifically, uncontrolled thyroid dysfunction, severe cardiovascular diseases such as heart failure or peripheral arterial disease, active autoimmune or rheumatic diseases, neurological disorders affecting autonomic function, as well as diabetes mellitus and active dermatological conditions in the facial area) were systematically excluded from the investigation [42]. The study population was consequently restricted to healthy volunteers devoid of any concomitant chronic pathologies that might potentially confound the experimental measurements. A combined methodology of visual inspection and palpation was employed to assess facial SAT development. Visual examination was conducted under standardized, uniform lighting conditions (consistent, diffuse overhead illumination of 500–600 lux at the examination level, avoiding shadows and direct light sources that could cause glare or specular reflection on the skin surface) with participants in a seated position (Figure 4) [43].

The physician evaluated the severity and uniformity of facial soft tissues in typical areas of subcutaneous adipose tissue accumulation—specifically, the cheeks, zygomatic region, and chin. Palpation was performed by gently compressing the skin and subcutaneous layer between the thumb and index finger. The central part of the cheek was selected for assessment. The thickness of the soft tissue fold, its elasticity, and resistance upon compression were determined (Figure 5). The findings were compared with average anatomical norms [44]. A two-stage classification system for assessing was applied. Mild subcutaneous adipose tissue deposition is characterized by a thin, poorly defined fold with minimal soft tissue volume. Upon compression, the skin barely forms a noticeable thickness. Pronounced subcutaneous adipose tissue deposition is characterized by a thick, clearly visible skin-fat fold with abundant soft tissue volume. The fold easily forms upon compression. To minimize subjective variability, all assessments were conducted by a single specialist.

The measurement was performed using a manual caliper in the submental fat pad using a technology like traditional caliperometry. The bony edge of the lower jaw and chin were determined by palpation. Then, using the thumb and index finger of one hand, the skin fold together with the subcutaneous fat was gently grasped and slightly pulled downwards. Holding the caliper in the other hand, the ends of the caliper were placed in the middle between the base and the top of the fold, slowly pressing the thumb on the caliper platform until it clicked. After this, measurements were taken (Figure 6).

The division by the level of deficit or excess was performed based on the results of measuring subcutaneous fat on the face. Thus, a deficit (4.5 ± 1) was determined in 15 patients, of which 14 had BMI within the normal range and 1 patient had Underweight (Mild thinness); excess (15 ± 1) was also found in 15 patients, of which 10 had normal BMI and 5 Overweight (Pre-obese).

The categories were as follows: Lack of SAT—fold thickness <5 mm (significantly below normal, indicating subnormal fat deposits); Normal SAT—5–12 mm, which corresponds to the mean value (~9 mm ± 4 mm) in the population of adolescents and adults [45]; Excessive SAT—>12 mm, significantly exceeding the mean and standard deviation. These thresholds are based on empirical data: the average fold thickness in adults is 9 mm (SD ≈ 4 mm) [46].

The experimental protocol comprised multiple sequential phases (Figure 7).

The measurement was performed using a manual caliper in the submental fat pad using a technology like traditional caliperometry. The bony edge of the lower jaw and chin were determined by palpation. Then, using the thumb and index finger of one hand, the skin fold together with the subcutaneous fat was gently grasped and slightly pulled downwards. Holding the caliper in the other hand, the ends of the caliper were placed in the middle between the base and the top of the fold, slowly pressing the thumb on the caliper platform until it clicked. After this, measurements were taken.

The research protocol was developed in accordance with the technical characteristics of the device, the instructions for use and the recommendations of the International Academy of Clinical Thermology (IACT) [47]. To ensure sufficient resolution of the resulting image and accuracy of the results, the infrared thermal imaging camera was installed at 0.4 m from the subject’s face. Before use, the device was calibrated using a black body. The initial, post-injection and delayed-for-20 min measurements of facial skin temperature were carried out at the same point to avoid errors.

Prior to any therapeutic interventions, the baseline facial skin temperature was measured using infrared thermography. All measurements were conducted in a controlled environment with stable ambient temperature and minimal air turbulence to eliminate confounding variables. The results of IRT are influenced by the temperature of the surrounding air and the background [48]. To evaluate only changes in skin temperature, it is necessary to minimize the movement of air currents and create a comfortable temperature and humidity in the treatment room. Air flows help to remove heat from any object, including human skin. In conditions of uncomfortable temperature and humidity, the body can start compensatory processes of heat production or heat transfer to restore homeostasis, which can distort the results of the experiment. To achieve these conditions, air conditioning was carried out in the treatment room in advance with a comfortable temperature of 20 °C and humidity of 40% according to ISO 1:2022 standard. A few minutes before the start of the experiment, the air conditioning system was turned off, and windows and ventilation shafts were closed in the treatment room to eliminate unwanted air flows [49].

Following the initial thermographic assessment, the subject received targeted intradermal injections of the investigational compound at predefined anatomical sites. Post-procedural temperature monitoring was performed at two intervals: immediate post-injection phase, delayed phase (20 min post-administration). After the introduction of BTX-A into the muscle, it takes about 12 min for binding and 5 min for subsequent internalization of the molecules. Considering that BTX-A molecules are constantly washed out of the injection field with a half-life that can be minutes, and the process of penetration into nerve terminals takes approximately 17 min [2], we assumed that a period of 20 min would be sufficient for the penetration of the main part of the active substance and the beginning of its functioning. The acquired thermographic data were systematically recorded in a standardized datasheet for subsequent statistical analysis.

### Analysis of Quantitative Metrics

Quantitative parameters were assessed for normality using the Shapiro–Wilk test. Variables that conformed to a normal distribution were described using arithmetic means (M) and standard deviations (SD). The representativeness of the mean values is expressed using 95% confidence intervals (95% CI).

For quantitative data that did not follow a normal distribution, descriptive statistics included the median (Me) as well as the lower and upper quartiles (Q1–Q3). Categorical variables are presented as absolute frequencies and percentages, with 95% confidence intervals (CIs) for proportions calculated using the Clopper–Pearson method.

A comparison of the two groups based on a quantitative indicator, whose distribution in each group followed a normal distribution, was conducted using Student’s *t*-test under the assumption of equal variances, and Welch’s *t*-test in case of unequal variances. The comparison of proportional values in the analysis of 2 × 2 contingency tables was performed using Fisher’s exact test when the minimum expected frequency was less than 10.

As a quantitative measure of effect when comparing relative indicators, the odds ratio with a 95% confidence interval (OR; 95% CI) was calculated. The comparison of proportional values in the analysis of multiway contingency tables was performed using Pearson’s chi-square test.

To compare three or more related groups with a normally distributed quantitative variable, one-way repeated-measures analysis of variance (ANOVA) was applied. The statistical significance of changes in the indicators over time was assessed using Fisher’s F-test. Post hoc analysis was conducted using paired Student’s *t*-tests with the Holm’s correction.

## Figures and Tables

**Figure 1 diagnostics-15-02519-f001:**
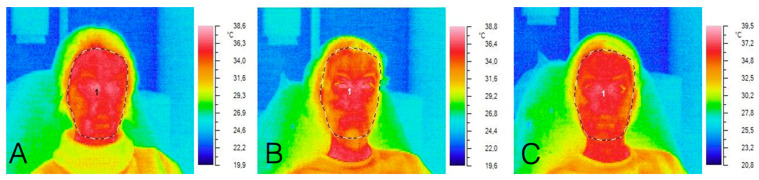
Thermograms of the subject: (**A**) before the procedure, (**B**) immediately after the procedure, (**C**) 20 min after the procedure.

**Figure 2 diagnostics-15-02519-f002:**
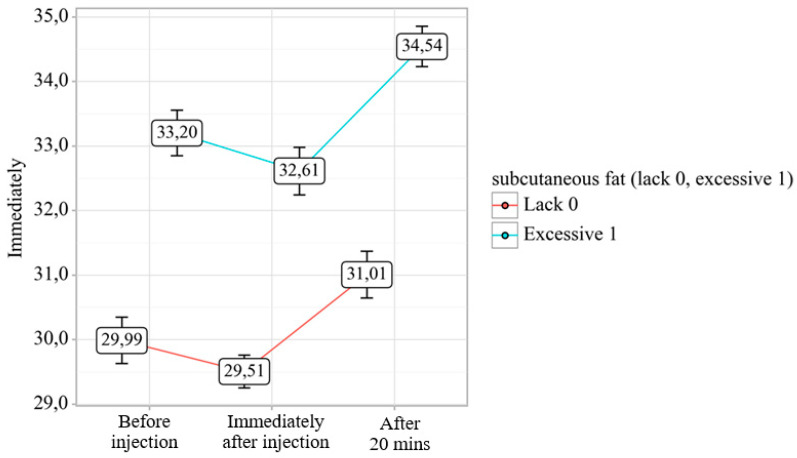
Analysis of temperature dynamics depending on the degree of subcutaneous fat development.

**Figure 3 diagnostics-15-02519-f003:**
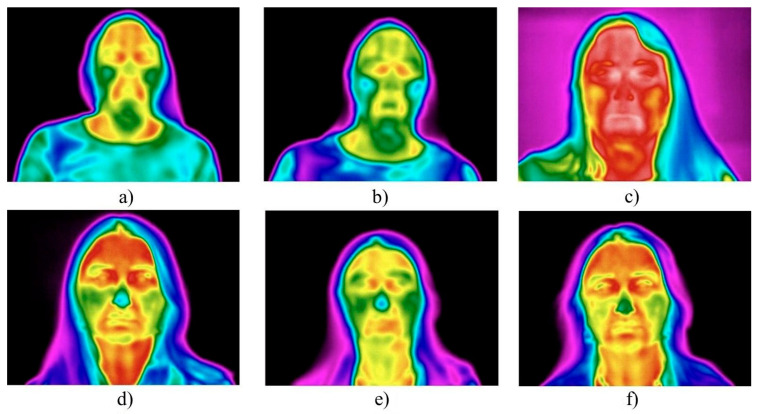
The effect of subcutaneous fat thickness on facial skin temperature post-procedure: (**a**–**c**) subjects with pronounced SAT exhibited more marked hyperthermia 20 min post-procedure; (**d**–**f**) subjects with minimal SAT showed milder temperature changes.

**Figure 4 diagnostics-15-02519-f004:**
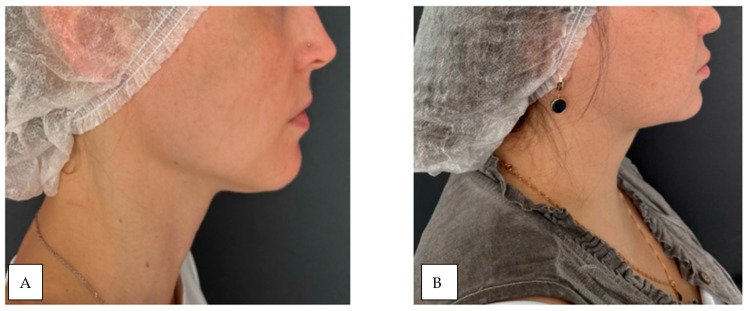
Demonstration of subcutaneous adipose tissue (SAT) expression variability: (**A**) minimal SAT deposition; (**B**) marked SAT accumulation.

**Figure 5 diagnostics-15-02519-f005:**
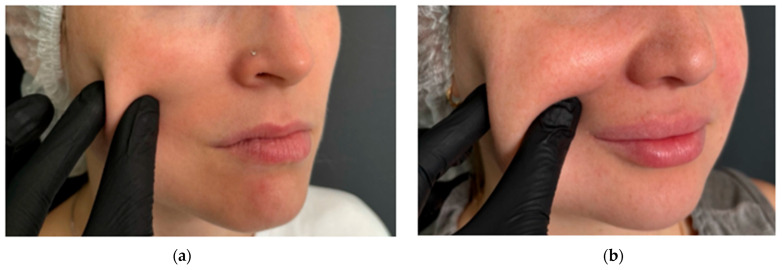
Assessment of soft tissue fold thickness in study participants: (**a**) minimal soft tissue fold expression in the midfacial region; (**b**) marked soft tissue fold prominence in the midfacial region.

**Figure 6 diagnostics-15-02519-f006:**
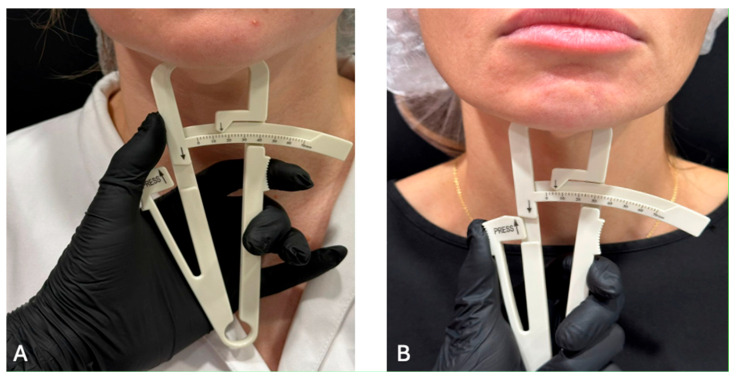
The measurement of the facial fat: (**A**) Excessive; (**B**) Lack.

**Figure 7 diagnostics-15-02519-f007:**
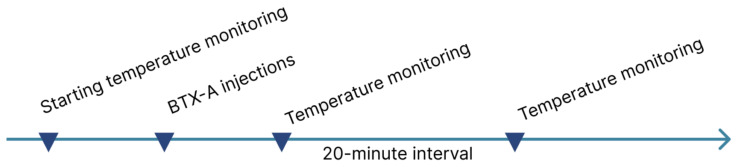
Phases of the experimental protocol.

**Table 1 diagnostics-15-02519-t001:** Temperature dynamics depending on subcutaneous fat thickness (Student’s *t*-test).

Subcutaneous Fat	Stages of Observation						*p*
	Before injection, °C		Immediately after injection, °C		After 20 min, °C		
	Me	Q_1_–Q_3_	Me	Q_1_–Q_3_	Me	Q_1_–Q_3_	
Lack (n = 15)	29.80	29.60–30.50	29.40	29.30–29.75	30.90	30.60–31.40	<0.001
	*p* Before injection–Immediately after injection = 0.035 *p* Before injection–After 20 min = 0.010 *p* Immediately after injection–After 20 min < 0.001						
Excessive (n = 15)	33.10	32.85–33.70	32.70	32.10–32.90	34.70	34.35–34.85	<0.001
*p*	<0.001		<0.001		<0.001		
	*p* Before injection–Immediately after injection = 0.021 *p* Before injection–After 20 min = 0.021 *p* Immediately after injection–After 20 min < 0.001						

**Table 2 diagnostics-15-02519-t002:** Analysis of BMI categories based on subcutaneous fat thickness.

BMI Range	Subcutaneous Fat		*p*
	Lack	Excessive	
Normal range	14 (93.3%)	10 (66.7%)	0.036 *
Overweight (Pre-obese)	0 (0.0%)	5 (33.3%)	
Underweight (Mild thinness)	1 (6.7%)	0 (0.0%)	

*—the observed differences were statistically significant (*p* < 0.05).

## Data Availability

Due to local law, we cannot make anonymized data publicly available, so all materials are kept by the authors and can be provided to readers upon request.

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
