# Peer review of "Evaluation of the Effectiveness of Botulinum Therapy Based on the Anthropometric Characteristics of the Face Using Non-Invasive Thermal Imaging Data"

_diagnostics, 2025, doi:10.3390/diagnostics15192519_

Round 1
Reviewer 1 Report
Comments and Suggestions for Authors
The authors present a cross-sectional observational study in which they aim to determine variations in facial skin temperature following the administration of botulinum toxin, measured using infrared thermography, with the aim of preventing asymmetries.
First, I would like to congratulate the authors for conducting clinical research focused on the prevention of unwanted adverse effects. It is essential that clinics have objective and reliable means of early detection of any iatrogenic effects resulting from invasive clinical procedures. This allows for the timely implementation of measures to mitigate these effects and enhance patient well-being and health outcomes. I encourage your team to continue this line of research, as it will benefit both clinicians and patients.
However, the submitted manuscript has several significant limitations that must be addressed to better understand the work carried out and the real scope of the conclusions.
The manuscript generally contains errors in the presentation of the content and requires significant improvements in the methodology section. I will specify these improvements in my recommendations.
I would like to clarify my position to the authors. My contributions are intended to ensure that their manuscript meets the quality standards they have achieved in their research work. However, these standards are not sufficiently reflected in the manuscript.
I will begin with a point that appears at the end of the text, but which is important to comment on first. The protocol was approved in 2021, and the authors must justify why they have used a 4-year-old protocol for a study supposedly conducted in 2025.
Turning to the "Introduction" section, the relationship between changes in skin temperature and the appearance of asymmetries remains to be fully elucidated. However, this information appears in the first part of the "Discussion" section. I recommend that the authors review the text between lines 133-174 and move the more general aspects related to physiology, which explain the proposed relationship, to the "Introduction" section.
Continuing in the "Introduction" section, the objective of the research does not align with the purpose of the work. Neither the botulinum toxin intervention nor its relationship with facial asymmetry is addressed in the text.
Turning to the "Results" section, it is important to note that the "Practical recommendations" subsection is not applicable here. I recommend moving it to the "Discussion" section.
In the "Discussion" section, in addition to the note made to move much of the initial content to the introduction, I recommend that lines 174-178 provide a more detailed explanation of the experimental and theoretical data referred to by the authors, as well as adding the appropriate bibliographic references on which their argument is based.
Finally, I will detail my recommendations for the "Materials and Methods" section, which requires extensive work to improve the manuscript. I will enumerate them below:
1. As indicated on line 218, it is essential to cite prior studies involving the K-MED device, which provide substantiation for its utilization.
2. Please ensure that the statistical software program is correctly named on line 223.
3. In line 224, please either reference the sample size calculation or studies in the same field that have used a similar sample. This is done to validate the sample obtained.
4. On line 228, please specify the chronic diseases in question and provide a rationale for their exclusion.
5. In line 231, please explain how a chronic disease can be a confounding variable in this study.
6. In line 233, please provide a detailed definition of "optimal lighting conditions" or cite a relevant bibliography that would explain these conditions for this type of study.
7. On line 244, please include references to the technique for assessing skin fold elasticity and resistance, and provide a discussion of its validity.
8. In line 245, please provide bibliographic references for the "anatomical standards" mentioned by the authors.
9. A visual diagram should be included to clarify the "multiple sequential phases" mentioned by the authors in line 255.
10. Please note that the infrared imaging protocol used and the bibliographic references that validate it are missing on lines 255-256. It is essential to specify which ROIs were used for baseline imaging and post-intervention imaging.
11. Line 258: According to the text, a series of interventions were carried out to minimize variables related to temperature and air turbulence. These interventions should be thoroughly explained in the text and supported by bibliographic references that detail their application and justification.
12. Line 262 should provide a rationale for why images were taken 20 minutes after inoculation instead of at a different time, such as 10, 15, or 30 minutes. This explanation should be supported by the relevant bibliographic references.
It is my hope that these comments will assist the authors in refining their manuscript and showcasing the effort and quality of their work in their field of research.
Author Response
Hello, dear reviewer!
On behalf of the authors, I express my deep gratitude to you for the time you devoted to our article and for the valuable comments that helped us take a fresh look at the results, improve the clarity and informativeness of the manuscript.
Comment 1: In relation to the conclusion of the ethical committee, we made a mistake, in fact from 2024, we mixed up the date
Comment 2: We have transferred the text in accordance with your recommendations, and also included 53-63 in it.
Comment 3: it is necessary to clarify the terminology: we were talking about temperature asymmetry (the blood flow rate in the microcirculatory bed on different sides of the body is different), and not about muscle dysfunction
Comment 4: Thank you for your advice, we have adapted our results to the findings at the lines 279-296
Comment 5: we made corrections at the lines 311-312
Comment 6: software program was named at the line 317
Comment 7: we have added our sample size calculation at the lines 319-323
Comment 8 and 9: united answer at the lines 324-330
Comment 10: optimal lighting conditions were discussed at the lines 334-338
Comment 11: anatomical standards are described at the lines 348-349
Comment 12: Phases of the experimental protocol and figure are at the 381-382 lines
Comment 13: we've added reference for imaging protocol at the lines 395-401
Comment 14: about minimizing variables related to temperature and air turbulence we wrote at the lines 405-416
Comment 15: time choice was described at the lines 420-426
Reviewer 2 Report
Comments and Suggestions for Authors
This manuscript raises a potentially interesting research question, but the rationale, mechanistic explanations, and methodological clarity require substantial strengthening. Addressing the above points will significantly improve the quality and interpretability of the work.
Major Comments
- Introduction – rationale
- The introduction should clearly articulate why botulinum toxin would be expected to induce changes in body temperature and describe the putative physiological mechanisms. At present, the rationale is missing.
- Study objective
- The objective is defined as evaluating the relationship between the hyperthermic response of facial vascular networks and the degree of subcutaneous adipose tissue (SAT) thickness. However, no explanation is provided of how SAT thickness could plausibly influence cutaneous vascular reactivity. Without this, the rationale and purpose of the study are unclear to readers.
- Mechanism of biphasic thermal response
- The biphasic thermographic response is attributed to “toxin-induced vasodilation” and inflammatory mediator release. This interpretation is problematic: while the molecular action of botulinum toxin is rapid, the clinical onset is delayed. The authors need to demonstrate direct vascular effects before attributing such rapid changes to the toxin.
- Definition of vascular response parameters
- The manuscript must define precisely what is meant by “velocity” and “magnitude” of vascular responses as measured in thermograms.
- Table 2 – SAT classification
- The categorization of SAT as “Lack” or “Excessive” is vague. Please provide explicit criteria (e.g., quantitative thresholds, measurement methods).
- Lines 133–137 – gamma motor neurons
- The statement regarding presynaptic inhibition of gamma motor neurons on intrafusal fibers is questionable.
- Gamma blockade is unlikely to occur immediately (within 20 minutes) after injection.
- The connection between gamma motoneurons and autonomic vascular terminals is not established.
- Please find and report evidence of spindles in frontal muscles
- Historical reference
- To my knowledge, the first to propose the use of botulinum toxin A for wrinkle reduction were Carruthers and Carruthers (1992).
- Line 152
- The statements here are imprecise and inaccurate; revision is required.
- Line 160 – SNAP-25 cleavage
- The claim that ~3% SNAP-25 cleavage is required for paralysis must be supported with references. Studies report that 80–90% cleavage is needed, and <50% is insufficient. Please cite appropriate sources.
- Early vascular effects (within 20 minutes)
- The reviewer considers it highly improbable that these are mediated by the toxin. More plausible explanations may include excipients, pH, osmolarity, or injection temperature—all of which are not reported in the methods. Please discuss.
- Injection-related effects
- The possibility that the injection procedure itself induces release of algesic/pro-inflammatory mediators is a reasonable explanation for immediate changes. However, persistence at 20 minutes is less convincingly explained. Please expand discussion.
- Clinical relevance of small temperature changes
- The manuscript should comment on whether the modest temperature reductions observed in the biphasic response have any clinical significance.
- Hypothermia and toxin uptake
- The claim that hypothermia reduces toxin uptake via vasoconstriction is interesting, but no supporting studies are cited beyond Raynaud’s phenomenon. Please provide additional references.
- Quantification of SAT
- The use of quantitative plicometry (or another validated method) would strengthen the methodology. At minimum, this limitation should be acknowledged.
Minor Comments
- Some sentences lack clarity and could be rewritten for precision (e.g., lines 152–160).
- Terminology should be standardized (e.g., SAT consistently defined).
- Ensure references are formatted according to journal guidelines.
Author Response
Hello, dear reviewer!
On behalf of the authors, I express my deep gratitude to you for the time you devoted to our article and for the valuable comments that helped us take a fresh look at the results, improve the clarity and informativeness of the manuscript.
Comment 1: at the lines 78-88 we write the rationale for why botulinum toxin would be expected to induce changes in body temperature
Comment 2: In lines 89-91 we write the rationale for why SAT thickness could plausibly influence cutaneous vascular reactivity
Comment 3: In lines 142-143 we demonstrate direct vascular effects of the mechanism of biphasic thermal response
Comment 4: Thank you for your question!
We mean the following phenomena (lines 113-116): the rate of vascular reaction is a time parameter reflecting how quickly the temperature changes after a functional test. And the magnitude of the vascular reaction is an amplitude parameter showing the intensity of the temperature change caused by shifts in blood flow.
Comment 5: For the categorization of SAT, we added a rationale for ranking subcutaneous fat volume using caliperometry (lines 334-348)
Comment 6: statement regarding presynaptic inhibition of gamma motor neurons on intrafusal fibers was supplemented and explained on lines 162-167 of the text
Comment 7: Historical reference has been corrected and added (number 15)
Comment 8: information about intravascular pH was added on lines 198-202
Comment 9: In response to your request, we have additionally clarified the data in the literature and supplemented the manuscript on lines 215-224.
Comment 10: Early vascular effects (within 20 minutes) were explained on lines 225-231
Comment 11: Injection-related effects were explained on lines 251-261, and the vegetomodulatory effect was also described.
Comment 12: We have long explained Clinical relevance of small temperature changes on lines 153-161 of the manuscript
Comment 13: We thank you for your comment. We confirm that toxin tolerance and uptake may indeed be reduced by hypothermia due to decreased blood flow and vascular permeability, and have corrected the manuscript at lines 173-184.
Comment 14: Thank you for your comment. In the revision of the manuscript, we clarified the classification criteria for the submandibular (SAT) on the face used in caliper measurements on lines 370-375.
Comment 15: We thank the reviewer for his valuable comments. We have rewritten this passage to improve its clarity and precision. The revised version appears below at lines 188-198.
Comment 16: Terminology has been standardized, the list of references has been brought into line with the journal's guidelines
Round 2
Reviewer 1 Report
Comments and Suggestions for Authors
Dear authors,
I would like to express my gratitude for your consideration of all the comments that have been submitted. It is to be commended that the manuscript has undergone significant improvement.
Author Response
Dear reviewer!
On behalf of the authors, I express my appreciation and gratitude for your attention and valuable comments, which have helped us improve our study.
Reviewer 2 Report
Comments and Suggestions for Authors
The manuscript has improved substantially; however, several aspects could still benefit from further refinement.
First, the introduction does not yet make sufficiently clear why botulinum toxin would be expected to exert a vascular effect—and consequently an effect on temperature—within such a short time frame. This rationale needs to be explicitly developed at the outset.
Second, some of the explanations currently placed in the Methods and Discussion sections would be more appropriately relocated to the Introduction, where they could help frame the research question and strengthen the overall rationale.
Third, the Discussion would gain in clarity if it began with a concise descriptive summary of the main results observed in this study. This is particularly important for a journal such as this one, where the Methods are presented at the end of the paper, and readers will benefit from having the findings clearly restated before their interpretation is attempted.
With regard to the vascular effects of botulinum toxin, the manuscript cites a study conducted on skin flaps. It would be more appropriate to cite work specifically addressing the autonomic or vascular system, ideally providing direct information on the time course of action and the onset of therapeutic effects in terms of autonomic response. It is well established that, in the muscular system, the earliest onset of botulinum toxin action typically occurs no sooner than 24–36 hours post-injection. For the claims in this manuscript to be persuasive, it is therefore essential to reference and discuss in detail studies or experiments that demonstrate a mechanism consistent with such a rapid onset of action at the vascular or autonomic level.
I also appreciated the inclusion of the paragraph on the possible presence of proprioceptors in facial muscles. However, what is missing is a logical link to the idea that basal activation of these muscles could represent a reflexive activity triggered by such sensors. This connection would enrich the discussion and place the hypothesis into a more coherent physiological framework.
Finally, in lines 213–228, the cited studies require more careful description in order to allow the reader to evaluate their reliability. The first citation specifies the type of muscle studied, the second refers generically to paralysis, and the third concerns miniature potentials. It is evident that these studies are not directly comparable in terms of scale or relevance, and this discrepancy should be explicitly acknowledged and commented upon.
Author Response
Dear Reviewer!
We thank you for your valuable comments, which have improved the quality of our manuscript and made the results more visual and the explanations more understandable to readers.
Comment 1.
We thank the reviewer for this important comment. We have added a rationale for the vascular reaction on lines 87-95.
Comment 2.
Your comment is valid, and we have updated the text in accordance with the recommendations on lines 105-109.
Comment 3.
We are grateful to the reviewer for this important clarification, as the order of the journal sections differs slightly from others. To improve the clarity and informativeness of the manuscript, we have made the necessary corrections on lines 152-155.
Comment 4.
We have clarified the order of temperature changes in the literature, and have amended the manuscript accordingly on lines 296-299.
Comment 5
Thank you for your comment. Indeed, in the previous version of the manuscript, the connection between basal activation of these muscles and reflexive activity triggered by such sensors was not clear to the reader, so we have made the necessary corrections on lines 176-184.
Comment 6.
You are correct, and in the previously provided citation, some inconsistency and unevenness in the presentation of the data was noted. We have brought this fragment into the order of logical presentation of the text on lines 227-260
Round 3
Reviewer 2 Report
Comments and Suggestions for Authors
paragraph 105-109 needs citations.
line 183 strabismus was the first medical condition treated with BoNT
paragraph 249-260 I do not understand the point...
Author Response
Dear Reviewer,
On behalf of my co-authors, we thank you for the time and attention you have given to our manuscript and for your valuable comments. This collaborative effort has highlighted the significance of focusing on detail.
Comment 1: The suggested references have been included on lines 105-109.
Comment 2: We appreciate this pertinent remark. Corrections have been applied to lines 181-182 to ensure historical precision.
Comment 3: We thank you for this comment. The paragraph in question has been rewritten, and the required changes have been made to lines 252-258.
